# Physical Activity Practice and Optimal Development of Postural Control in School Children: Are They Related?

**DOI:** 10.3390/jcm9092919

**Published:** 2020-09-10

**Authors:** Jose L. García-Soidán, Jesús García-Liñeira, Raquel Leirós-Rodríguez, Anxela Soto-Rodríguez

**Affiliations:** 1Faculty of Education and Sport Sciences, University of Vigo, Campus a Xunqueira, s/n. 36005 Pontevedra, Spain; jlsoidan@uvigo.es (J.L.G.-S.); jesgarcia@alumnos.uvigo.es (J.G.-L.); 2Nursing and Physical Therapy Department, Faculty of Health Sciences, Campus Ponferrada, University of León, 2440 Ponferrada, Spain; 3Health Service from Galicia (SERGAS), Galician Health Services—Ourense Hospital, s/n. 32005 Ourense, Spain; anxelasoro@hotmail.es

**Keywords:** postural balance, exercise, child development, sex characteristics

## Abstract

Background: This study aims to analyze the effect of physical activity practice on the postural control state of school children. If such an effect was detected, the second aim of the study was to identify which specific capacities of postural control benefited the most from physical activity. Methods: A cross-sectional study was performed using a convenience sample of 118 healthy children (54 girls) with a mean age of 10.3 ± 1.2 years. Their weight and height were measured. The accelerometric assessment of balance included four different tests in static balance and walking. Results: Physical activity habit prevalence was 38.9% in girls and 60.9% in boys, and its frequency was 2.3 days per week in girls and 2.8 days in boys. The active children obtained lower accelerations, but the active and sedentary girls showed lower accelerometric values than the active boys. The logistic regression analysis demonstrated the influence of sex on the accelerations of the body (*p* < 0.001), regardless of the habit of physical activity. Conclusions: Active children have better postural control than sedentary children, although sedentary girls have better balance than active boys. Therefore, physical activity practice seems to favor a more efficient development of postural control, but it cannot level or reverse the effect of the neurophysiological factors that are conditioned by sex.

## 1. Introduction

The optimal development of psychomotor skills and capacities, such as postural control, occurs during early childhood, and it is necessary, once in adulthood, for the attainment of the optimal cognitive and psycho-social functioning [1] and other more complex psychomotor abilities [2]. In early childhood, especially between 6 and 12 years of age [3], postural control develops and reorganizes permanently, adapting, through experience, to the constant growth of the body and to the incorporation and improvement of biomechanical strategies, such as anticipatory postural adjustments and alignment and balancing reactions [4,5,6].

Previous research has confirmed the differences between girls and boys in the maturation process of the central nervous system. Girls are often better at skills that require balance and precision (such as walking on tiptoe, walking on a balance beam, walking in tandem, or staying monopod) [7,8,9]. However, children often achieve better results in activities that require speed and strength [10,11]. As a possible contextual explanation for these phenomena, Webster et al. [12] confirmed that boys participated more in total physical activity (PA) and moderate–vigorous PA, and they participated less in sedentary behavior.

PA involves a series of physiological reactions. One of the most relevant of such reactions is the secretion of neurotransmitters and hormones that facilitate intellectual development, as they enhance the generation of new neurons and multiply and strengthen the existing neural connections among the brain areas related to memory and learning [13,14]. These mechanisms, which are present throughout the entire life of an individual, are especially relevant in children, as PA improves their basic intellectual capacities, such as emotional control, memory, and the capacity to adapt to different tasks and environments, which are essential for an optimal psychological and physical development [15,16].

One of the physical aptitudes that can benefit from PA is postural control, which is defined as the capacity of an individual to maintain the centre of gravity of his/her body over his/her base of support and against gravity [17]. Postural control is a system that uses and coordinates different strategies: biomechanical (especially in the lower limbs), movement (through unconscious postural adjustments), sensory (using somatosensory, visual, and vestibular information), and cognitive (through attention and learning from experiences) [18,19,20,21]. However, balance assessments are usually based on qualitative methods, which are inefficient and have low reliability [22]. Reliable tests have been developed in a limited way, but they require the use of an expensive force or pressure platforms, magnetic tracking, infrared emitter, electronic pressure sensitive walkway, or surface electromyographic recordings to determine the individual’s center of pressure. This is a very reliable and valuable clinical indicator to identify relatively premature sensory-motor deficits. On the contrary, numerous studies have been performed to assess equilibrium in reference to the global behaviour of the individual and not specifically their centre of mass [23,24]. Accelerometry allows the analysis of specific aspects related to this factor, and it has been proposed as a new implementation method due to the inexpensiveness, reliability, portability, and comfortability for the evaluator [25].

The exact mechanism through which PA influences the development of alignment and balancing reactions in preschool and school children has still not been established using a reliable, sensitive, and specific system such as accelerometry [23]. Consequently, this study was carried out with the aim of analysing the effect of PA on the postural control state of school children. Furthermore, if such effect was detected, the second aim of the study was to identify which specific capacities of postural control benefited the most from PA in children, and finally, to identify the possible differences between both sexes. The initial hypotheses were that PA practice influences the maturation process of the postural control system and this influence is not the same in boys and girls.

## 2. Materials and Methods

### 2.1. Study Design and Sample

This descriptive, cross-sectional study was performed using a convenience sample of children from Pontevedra (Spain). All participants were recruited from their school. Participants who met any of the following exclusion criteria were unable to participate: children who were unable to walk independently or without external orthotics, those who could not stand for 60 s or more, children with stage 5 of biological development according to Tanner, children with any specific contraindications to the evaluation tests; or children who did not provide parental informed consent to participate in the study.

This study involved 118 healthy children (54 girls), with a chronological mean age of 10.6 ± 0.9 years old (range of 8–12 years). For maturational stage, Tanner’s self-evaluation method of pubertal maturation was used [26], which consists of choosing the stage that best suited each child, based on one of the images on a card that ranged from grade one to five (1 = pre-puberty; 2–4 = puberty; 5 = post-puberty). This is a non-invasive, reliable, and valid indicator of biological maturation [27,28].

### 2.2. Procedure

All procedures performed in this study were in accordance with the ethical standards of the Commission of Ethics of the Faculty of Sciences of Education and Sport of the University of Vigo (Pontevedra, Spain; number 3-0406-14). The parents of all participants received an explanation about the objective of the study and the evaluations that were going to be carried out; later, all signed an informed consent form in accordance with the Declaration of Helsinki (revised 2013).

First, the age of the children and whether they practiced PA were recorded using a dichotomous question: did they practice PA (active) or not (sedentary) and, if so, the number of days they did it was also noted. The parents were asked the following question: “PA is defined as any bodily movement produced by skeletal muscles that results in energy expenditure and is a behavior that results in an elevation of energy expenditure above resting levels. Considering the definition above, does your child participate in an activity, at least once a week and for at least 30 min, that includes these characteristics?” [29,30].

Then, their weight and height were measured (using a SECA^®^ scale and stadiometer, Berlin, Germany). For both anthropometric measurements, students were asked to remove their shoes and any unnecessary clothing.

After these measurements, an accelerometer was placed in the lumbar medial zone (coincident with the fourth lumbar vertebra). This specific location was the 4th lumbar vertebra that has been demonstrated to reflect the behaviour of the center of mass [23,24]. The devices were attached with adhesive tape to avoid displacement. The trial was explained to the participants, and they were accompanied to the corresponding measurement room for testing.

The accelerometric assessment included four different tests: (a) static balance on one leg with eyes closed (OLCE); (b) static balance on one leg with eyes open (OLOE); (c) balancing on one leg on a foam mat with eyes open to induce the appearance of dynamic balance reactions (DOL); and (d) walk at normal speed towards a cone located 10 m away, with each participant walking around the cone and returning to the starting point (NG). The OLCE, OLOE, and DOL trials had a fixed duration of 30 s. The duration of the NG trial varied depending on the time required by the participant to finish the circuit. Each test was separated at 30 s intervals from the next to prevent the effects of lower extremity muscle fatigue [23,31]. For the DOL test, a mat with a density of 30 kg/m3 was used as a support surface on the ground. Participants were told that if they suffered an imbalance while in a monopodal stance that required them to use their other leg for support, they should attempt to recover the requested position in the shortest time possible. During testing, the subjects wore socks and comfortable clothing so that they could perform the tests comfortably. In the monopodal balance tests, the participants chose the leg on which to make the support. For that, they were allowed to make previous attempts to make the selection (which they had to respect for all the tests).

The sequence of the trials was OLCE–NG–OLOE–DOL–OLCE–DOL–NG–OLOE–DOL–NG–OLOE–OLCE, with each trial (OLCE, OLOE, DOL, and NG) performed three times.

### 2.3. Instrument and Processing of Data

The accelerometer GT3X+ (Actigraph^®^, Pensacola, FL, USA) provides data on body movements in three axes: the vertical axis (from the transverse plane); sagittal axis (coronal/frontal plane); and the perpendicular axis (anteroposterior plane). In addition, the root mean square (RMS) was used as representative value of the three-axis module vector. The accelerometer measurements were configured for a time frame of 1 s. This protocol has already been previously applied and validated both in the adult population and in children [23,32,33].

Using a low frequency helps to eliminate any acceleration noise [24]. Thus, a frequency of 50 Hz was selected to achieve greater accuracy in the analysis of postural balance [23]. Raw data were collected at the selected sample rate and processed using ActiLife software. The accelerometers used in the tests were synchronised using a PC and ActiLife software. The highest accelerometric value in each attempt of each test was selected, and the maximum value of the three repetitions was averaged.

### 2.4. Statistical Analysis

A descriptive analysis of all the study variables was performed through the calculation of the average values (to determine the central tendency) and standard deviation (as a measure of dispersion).

The variables showed a normal distribution according to the Kolgomorov–Smirnov test (*p* > 0.05), and there was a homogeneity of variances, applying the Levene test. The *t*-test was used to verify the existence of significant differences between the sexes and ANOVA test with the Bonferroni correction to do it between the different groups of physical activity and sedentary lifestyle divided by sex.

A correlation analysis was performed between the PA practice and the accelerometric variables to find the relationship between them. The correlation between the different accelerometric variables was also found to determine the relationship between the accelerations produced and the different balance tests performed.

We applied the logistic regression model (logit) to analyse the association of independent variables and the dependent dichotomous variable (sex: 0 = boy; 1 = girl). Finally, adjusted odds ratios (OR) with their confidence intervals were estimated using the multivariate regression model. The model was initially adjusted by age and PA practice. The program Stata version 13 (Stata Corp., College Station, TX, USA) was used for statistical analysis.

## 3. Results

### 3.1. Descriptive Analysis

The results indicate that there were no significant differences between sexes in terms of age, height, weight, or BMI (Table 1). In contrast, the biological age was significantly higher in girls (Table 1 and Table 2). Regarding PA, there were differences between boys and girls in habit prevalence (38.9% of active girls vs. 60.9% of active boys) and frequency (2.3 days per week in girls vs. 2.8 days per week in boys).

With respect to accelerometry, the results showed that the increasing complexity of the tests induced accelerations of proportionally increasing magnitude: the lowest accelerations were detected during the OLOE test, followed by OLCE and DOL, and the highest accelerations were obtained in the NG test. Furthermore, during the two tests in monopodal stance, the lowest and highest accelerations were observed in the vertical axis and in the mediolateral axis, respectively. In the test conducted on a mat, the highest accelerations were also obtained in the mediolateral axis, although the lowest accelerations were observed in the anteroposterior axis, as in the NG test, where the highest accelerations were obtained in the vertical axis.

In the comparison between sexes, all the accelerometric variables of the tests in monopodal stance were statistically lower in the female group (Table 1). However, in the gait test, although the accelerations were also lower in the girls, they were only statistically significant in the mediolateral axis and in the RMS of the accelerations. 

### 3.2. Effect of PA Practice on Accelerations

The active children obtained lower accelerations, except in the mediolateral axis in OLCE and NG. The comparative analysis between sedentary and active children by sex showed similar results, with some exceptions: the active boys only obtained higher accelerations in the mediolateral axis in OLCE. Moreover, the active girls obtained higher accelerations than the sedentary girls in the gait test (Table 3). Of all these differences, the only one that showed statistical significance was that between the active and sedentary girls for the mediolateral and anterioposterior axes in DOL.

However, in the comparison between the active and sedentary subgroups according to sex, the active girls showed lower accelerometric values than the active boys, except in the vertical axis in NG. These values were statistically lower in the mediolateral axis in the three tests in monopodal stance, in the vertical axis in OLCE and DOL, and in the anteroposterior axis and RMS only in DOL.

Lastly, comparing between sedentary boys and girls, the boys obtained significantly higher accelerations in all axes in all four tests, except in the anteroposterior axis in NG.

### 3.3. Correlation Analysis

The correlation analysis showed an inverse relationship between chronological age and the magnitude of the accelerations in the RMS and in the mediolateral and anteroposterior axes in OLOE (r = −0.05; *p* < 0.001) and between chronological age and the magnitude of the accelerations in the mediolateral axis in DOL (r = −0.4; *p* < 0.001). On the contrary, the biological age did not obtain any significant correlation with the accelerometric variables.

Among the recorded accelerations, the values of RMS and of the mediolateral and anteroposterior axes in NG were correlated with those of the four values of OLOE (r = 0.5; *p* < 0.001). The RMS and accelerations in the mediolateral axis in NG were correlated with the four accelerometric values of DOL (r = 0.4; *p* < 0.001). The number of days of PA practice were not correlated with any of the accelerometric variables.

### 3.4. Logistic Regression Analysis

The logistic regression analysis demonstrated the influence of sex on the accelerations of the body (Table 4). The four variables of the tests in monopodal stance revealed that their results were influenced by sex, especially for the mediolateral axis in OLOE and OLCE (marginal effect = −0.01; *p* < 0.001). None of the variables from NG obtained significant results. This analysis showed that the female participants obtained the lowest accelerations (0.003 < OR = 0.01, *p*-value < 0.05), even after adjusting for age and PA practice.

## 4. Discussion

The aim of this study was to analyse the effect of PA practice on the state of balance in children. If such an effect was detected, the second objective was to identify which specific capacities of postural control benefit the most from PA practice in children. The analysis of the results showed that PA practice influences postural control, although to a lesser extent than the biological factors that are conditioned by sex.

Firstly, in agreement with the findings reported in previous studies [34,35], the girls of the present investigation practised PA in a smaller number and less frequently than the boys. Considering this finding and the fact that the participants of this study were healthy boys and girls, the subsequent results are based on participants that meet the usual characteristics of the child population of developed Western countries.

The observed accelerations revealed that the tests induced increasingly greater alignment (trunk straightening reactions) and balancing reactions in proportion to the increasing complexity of the tests: the lowest accelerations were obtained in OLOE, followed by OLCE, DOL, and NG. Physiologically, this finding is consistent with the theory of motor control [36,37] and with the theory of postural control [38,39,40], considering that the positions with eyes open are less challenging for the participants than the tests with deprivation of visual information, on an unstable surface, and in motion (gait) [18]. Moreover, the highest accelerations in monopodal stance were obtained in the mediolateral axis, which is in line with the results of previous studies in both children [23] and older adults [25]. Sudden movements (i.e., higher accelerations) in the mediolateral axis have been strongly associated with the risk of falling [41], and maximum accelerations in such an axis in dynamic tests have been directly correlated with the strength of the lower limbs [32]. However, the highest accelerations in NG were found in the vertical axis, which is in agreement with the gait pattern reported in previous studies with adults and older adults [32]; that is, the fact that the highest accelerations were identified in the vertical axis and not in the mediolateral axis confirms the healthy state and normality of the participants, and that, at their age, the gait pattern is both fully established and equivalent to that of adults.

In the comparison between sexes, all the accelerometric variables of the tests in monopodal stance were statistically lower in the group of girls. This suggests that girls are able to face conditions that challenge postural control and overcome them by executing slower (i.e., less accelerated) movements than boys [23]. However, this finding may be due to the fact that the girls studied were in more advanced stages of maturational development. Prior studies have linked motor development in childhood to the growth of the central nervous system [42,43] and have assumed that age and sex relationships among biological maturation and fundamental motor skills (such as postural control) may be associated with changes in brain structure and underlying function neuromuscular maturation [44]. In parallel, Freitas et al. [42] indicate that taking into account the sex differences in biological maturation, skeletal maturation may be more closely related to neuromuscular maturation in girls than in boys at these young ages.

The active boys and girls obtained lower accelerations than the sedentary boys and girls, which confirms the beneficial effect of exercise on abilities such as balance. However, the active boys, despite the PA practice, showed much higher accelerations than the sedentary girls in the tests of static balance. This reveals that PA has beneficial effects on the state of postural control in both boys and girls, although with lesser influence on the quality and magnitude of alignment and balancing reactions than the biological factors that are conditioned by sex (such as neurophysiological, physical, hormonal, and sexual maturational differences) [40,45].

Furthermore, the active girls showed higher accelerations than the sedentary girls in the gait test, which was probably because since the former are physically more active, they could walk normally at a higher speed, resulting in higher accelerations. However, this cannot be confirmed, as the duration of the gait test was not recorded.

Of all these differences between active and sedentary children, statistical significance was only found in the comparison between girls in the mediolateral and anteroposterior axes in DOL. Therefore, the greatest benefits of PA practice for balance are generally observed when the postural control is challenged with a smaller base of support and on an unstable surface. Such assessment conditions disrupt the information provided by the somatosensory subsystem (which consists of the proprioceptive and tactile information pathways) [18,46,47] and by the vestibular subsystem (which sends information about the orientation of all the body parts with respect to gravity and the support surface) [18,48]. However, the differences between active and sedentary children were not significant in any case. This finding is novel and even incoherent, considering that the boys claimed to practise PA more frequently than the active girls. Furthermore, despite the possible variability in the characteristics of the PA practised by boys and girls, it is known that boys have a preference for team sports such as football and basketball [49], whose practice involves the strengthening of the muscles of the lower limbs, which is one of the indirect variables with the greatest capacity to improve the results of the balance assessment tests [3,50,51].

On the other hand, in the comparison between active boys and girls, the latter obtained even lower accelerations than the former, although the differences between sexes were considerably lower in the active children than in the sedentary children. Therefore, although PA did not level the results between sexes, it managed to reduce the differences between them in the sedentary children.

The correlation analysis revealed an inverse relationship between age and the magnitude of the accelerations in the monopodal tests with eyes open. This suggests that the maturation of the vestibular systems and brain centres of postural control occurs at a later stage, especially for the coordination between somatosensory and vestibular information, which is the one used preferentially in monopodal stance, both on the floor and on a mat.

The number of days of PA practice was not correlated with any of the accelerometric variables, which suggests that the adequate development of postural control does not require a minimal dosage of PA; this rather indicates that postural control benefits from the usual PA practice of children’s lifestyle, or at least two days per week, which was the usual dose of physical exercise of the participants.

Furthermore, the results of the logistic regression analysis demonstrated the influence of sex on the accelerations of the body, which is in line with the results of the previous analyses. The four variables of the test in monopodal stance revealed that its results were influenced by sex, especially in the mediolateral axis in OLOE and OLCE. None of the variables of NG obtained significant results in any of the statistical tests used. This could be because since walking is frequently required in numerous aspects of daily living, the gait test cannot be used as a discriminative or diagnostic tool in a sample of healthy children.

This study has some limitations that must be pointed out, such as the sample size, which is not large enough to generalise the results to the global population of children, and the shortage of data about PA practise (e.g., type and time since first practice). These aspects do not allow elaborating a more detailed theoretical justification of the obtained results. However, this is the first study to delve into the relationship between PA practice and its influence on postural control in a differentiated manner between boys and girls through the specific and sensitive analysis of accelerometry in static and walking balance tests. Another limitation is the difference in biological age detected between boys and girls (a deficiency that should be resolved by future research when comparing matched groups by maturational status).

As future research lines, it is necessary to conduct studies with larger sample sizes and a longitudinal follow-up along time to observe how the balancing reactions evolve in children depending on their PA practice.

## 5. Conclusions

Girls are able to face conditions that challenge postural control and overcome them by executing smoother (i.e., less accelerated) and slower movements than boys. That is, in general, they have better postural control than boys. Similarly, active children have better postural control than sedentary children, although sedentary girls have better balance than active boys. Therefore, PA practice could have the ability to favor a more efficient development of postural control, but it cannot level or reverse the effect of the neurophysiological factors that are conditioned by sex.

Thus, the sedentary lifestyle must be reduced in the children population, especially among girls. Moreover, the PA programmes for children must promote the incorporation of activities that stimulate the activity of the somatosensory system through exercises conducted under monopodal stance conditions with a deprivation of visual information and/or on unstable surfaces.

## Figures and Tables

**Table 1 jcm-09-02919-t001:** Descriptive analysis of the sample (mean ± standard deviation).

	All (*n* = 118)	Girls (*n* = 54)	Boys (*n* = 64)
Chronological age (years)	10.6 ± 0.9	10.7 ± 0.9	10.5 ± 1
Maturational Tanner’s stage	1.6 ± 0.6	1.3 ± 0.5 **	1.8 ± 0.7 **
Height (cm)	143 ± 0.1	146 ± 0.1	142 ± 0.1
Weight (kg)	40.1 ± 10.5	41.9 ± 11.1	38.5 ± 9.7
BMI (kg/m^2^)	19.1 ± 3.6	19.4 ± 3.5	18.9 ±3.6
Physical activity practice (days)	2.7 ± 0.9	2.3 ± 1.2 *	2.8 ± 0.6 *
**Accelerometric variables**			
Monopodal balance with eyes open (g)			
Vertical axis	21.4 ± 27	13.4 ± 15.1 **	28.1 ± 32.5 **
Mediolateral axis	31.3 ± 25.2	22.1 ± 16.8 ***	39 ± 28.5 ***
Anteroposterior axis	29.7 ± 27.1	23.8 ± 23.4 *	34.6 ± 29.2 *
Root mean square	47.8 ± 40.2	36.3 ± 29.9 **	57.5 ± 45.3 **
Monopodal balance with eyes closed (g)			
Vertical axis	37.2 ± 32.5	26.8 ± 23.6 ***	46 ± 36.5 ***
Mediolateral axis	50 ± 27.9	38.8 ± 21 ***	59.4 ± 29.7 ***
Anteroposterior axis	41.9 ± 26.6	34.6 ± 23.8 **	48 ± 27.6 **
Root mean square	72.1 ± 42.2	57.1 ± 32 ***	84.8 ± 45.8 ***
Monopodal balance on mat with eyes open (g)			
Vertical axis	47.9 ± 44.8	33.3 ± 35.2 ***	60.2 ± 48.6 ***
Mediolateral axis	52.4 ± 35.4	39.2 ± 27.4 ***	63.6 ± 37.6 ***
Anteroposterior axis	42.4 ± 31.7	33 ± 26 **	50.2 ± 37.6 **
Root mean square	80.9 ± 56.5	62.6 ± 46 ***	96.3 ± 60.4 ***
Normal gait (g)			
Vertical axis	64.3 ± 22.2	61.6 ± 15.4	66.7 ± 26.5
Mediolateral axis	59.3 ± 17.3	55.5 ± 15.3 *	62.4 ± 17.8 *
Anteroposterior axis	54.1 ± 19.7	51.4 ± 12.4	56.5 ± 24.2
Root mean square	92.5 ± 29.5	86.5 ± 20.1 *	97.6 ± 34.9 *

*t*-test between sex: * *p* < 0.05; ** *p* < 0.01; *** *p* < 0.001.

**Table 2 jcm-09-02919-t002:** Maturational distribution of participants.

Age (years)	Boys Development Stage			Girls Development Stage		
	1	2	3	1	2	3
8	5	-	-	2	-	-
9	15	2	-	3	3	-
10	12	4	-	12	9	2
11	11	12	-	1	13	5
12	1	2	-	-	2	2
All	44	20	-	18	27	9

**Table 3 jcm-09-02919-t003:** Comparative analysis between active and sedentary children (mean ± standard deviation).

	Girls		Boys		All	
	Active (*n* = 21)	Sedentary (*n* = 33)	Active (*n* = 39)	Sedentary (*n* = 25)	Active (*n* = 60)	Sedentary (*n* = 58)
**Monopodal balance with eyes open** (g)						
Vertical axis	12.1 ± 16.4	14.1 ± 14.9 ^###^	22.2 ± 29.1	37.3 ± 35.9 ^###^	18.7 ± 25.7	24.1 ± 28.3
Mediolateral axis	19.3 ± 17.1 ^&&^	23.9 ± 16.3 ^###^	35.5 ± 23.8 ^&&^	44.6 ± 34.4 ^###^	29.8 ± 22.9	32.8 ± 27.5
Anteroposterior axis	21.9 ± 26.4	25 ± 21.3 ^#^	27.8 ± 21.9	45.3 ± 35.9 ^#^	25.7 ± 23.5	33.7 ± 30
Root mean square	33.1 ± 34.8	38.2 ± 26.2 ^##^	50.3 ± 39.1	68.7 ± 52.6 ^##^	44.3 ± 38.2	51.4 ± 42.2
**Monopodal balance with eyes closed** (g)						
Vertical axis	24 ± 19.7 ^&^	28.6 ± 25.6 ^##^	41.9 ± 32.7 ^&^	52.5 ± 41.7 ^##^	35.6 ± 29.9	38.9 ± 35.2
Mediolateral axis	34.2 ± 23 ^&^	41.7 ± 19 ^##^	59.6 ± 31.1 ^&^	59.1 ± 27.9 ^##^	50.7 ± 30.9	49.2 ± 24.6
Anteroposterior axis	31.8 ± 20	36.4 ± 25.8 ^#^	44.5 ± 27.8	53.4 ± 26.9 ^#^	40.1 ± 25.9	43.7 ± 27.4
Root mean square	51.7 ± 30.3 ^&^	60.6 ± 32.6 ^##^	82.1 ± 46.8 ^&^	89.1 ± 44.8 ^##^	71.5 ± 44	72.9 ± 40.5
**Monopodal balance on mat with eyes open** (g)						
Vertical axis	29.9 ± 33.2 ^&^	35.4 ± 36.3 ^##^	56.4 ± 46.7 ^&^	66.2 ± 51.8 ^##^	47.1 ± 44.1	48.7 ± 45.9
Mediolateral axis	29.7 ± 23.6 **^;&&&^	45.2 ± 28.3 **^; #^	62.9 ± 34.8 ^&&&^	64.7 ± 42.3 ^#^	51.3 ± 35	53.6 ± 36
Anteroposterior axis	24.1 ± 18.8 *^;^^&&&^	38.7 ± 28.2 *^; #^	46.6 ± 31.7 ^&&&^	55.9 ± 37.9 ^#^	38.7 ± 29.7	46.1 ± 33.5
Root mean square	52 ± 40.1 ^&&^	69.3 ± 48.2 ^##^	91.7 ± 59.4 ^&&^	103.5 ± 62.5 ^##^	77.8 ± 56.4	84 ± 56.9
**Normal gait** (g)						
Vertical axis	63.2 ± 14.4	60.5 ± 16.1 ^#^	61.9 ± 23.7	74.1 ± 29.3 ^#^	62.3 ± 20.8	66.4 ± 23.5
Mediolateral axis	57.6 ± 16.7	54.3 ± 15.5 ^#^	61 ± 17.4	64.7 ± 18.6 ^#^	59.8 ± 17.1	58.8 ± 17.5
Anteroposterior axis	51.4 ± 13.5	51.4 ± 11.6	52.6 ± 17.6	62.5 ± 31.4	52.2 ± 16.2	56.2 ± 22.8
Root mean square	88.9 ± 20.1	84.9 ± 20.3 ^#^	91.4 ± 27.8	107.3 ± 42.5 ^#^	90.5 ± 25.2	94.5 ± 33.4

Comparison between active and sedentary girls: * *p* < 0.05; Comparison between active girls and active boys: ^&^
*p* < 0.05; ^&&^
*p* < 0.01; ^&&&^
*p* < 0.0001; Comparison between sedentary girls and sedentary boys: ^#^
*p* < 0.05; ^##^
*p* < 0.01; ^###^
*p* < 0.001.

**Table 4 jcm-09-02919-t004:** Logistic regressions for sex adjusted for age and physical activity practice.

	M. E.	O. R.	S. E.	95% C. I.
**Monopodal balance with eyes open** (g)				
Vertical axis	−0.008 **	0.968 **	0.011	0.948–0.989
Mediolateral axis	−0.01 ***	0.958 ***	0.011	0.937–0.981
Anteroposterior axis	−0.005 *	0.98 *	0.009	0.962–0.997
Root mean square	−0.005 **	0.981 **	0.006	0.968–0.994
**Monopodal balance with eyes closed** (g)				
Vertical axis	−0.005 **	0.977 **	0.007	0.962–0.991
Mediolateral axis	−0.01 ***	0.965 ***	0.009	0.946–0.983
Anteroposterior axis	−0.005 **	0.977 **	0.008	0.961–0.994
Root mean square	−0.005 ***	0.979 ***	0.006	0.968–0.991
**Monopodal balance on mat with eyes open** (g)				
Vertical axis	−0.004 **	0.983 **	0.005	0.973–0.994
Mediolateral axis	−0.006 ***	0.973 ***	0.007	0.959–0.987
Anteroposterior axis	−0.006 **	0.977 **	0.007	0.961–0.992
Root mean square	−0.003 ***	0.986 **	0.004	0.977–0.994
**Normal gait** (g)				
Vertical axis	−0.002	0.988	0.008	0.971–1.005
Mediolateral axis	−0.005	0.977	0.011	0.954–1.001
Anteroposterior axis	−0.227	0.985	0.01	0.964–1.007
Root mean square	−0.004	0.985	0.007	0.971–1.001

M. E.: marginal effects after logit in percentage; O. R.: odds ratio; S. E.: standard error; 95% C. I.: 95% confidence interval. * *p* < 0.05; ** *p* < 0.01; *** *p* < 0.001.

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
