# Peer review of "Physical Activity Practice and Optimal Development of Postural Control in School Children: Are They Related?"

_jcm, 2020, doi:10.3390/jcm9092919_

Round 1
Reviewer 1 Report
General comments
I appreciate the opportunity to provide comments on this manuscript, which sought to investigate the effect of PA on the postural control state of school children with two secondary objectives to identify which specific capacities of postural control benefited most from physical activity in children and to identify possible differences between the sexes. As I mentioned in my previous review, the paper is well written and I also find the thesis of this paper interesting and warranted but feel there are still issues to be addressed.
Specific Comments:
- Per my suggestion, the authors have included information concerning the chronological age of the participants. Further, tables 1 and 2 (newly created) were adjusted. However, these results were not addressed in a systematic manner in the discussion in the context of these differences. I mentioned the comments below but these were not addressed in the discussion or addressed to a minor degree.
- Lines 283-287: The authors mentioned: “This suggests that girls are able to face conditions that challenge postural control and overcome them by executing slower (i.e., less accelerated) movements than boys.” What was the reason for this finding? Since the mean chronological age of the participants in this study was higher, could this be related to differences in developmental age?
- Lines 291-294: A further and more detailed discussion is warranted.
Author Response
Dear Editor and Reviewer of Journal of Clinical Medicine:
Thank you very much for your suggestions and contributions to improve the quality of the manuscript. Following your indications, we respond, point by point, to the reviewers' comments.
In the text, all the modified or added sentences have been written in red to facilitate the correction by the reviewers.
- Per my suggestion, the authors have included information concerning the chronological age of the participants. Further, tables 1 and 2 (newly created) were adjusted. However, these results were not addressed in a systematic manner in the discussion in the context of these differences. I mentioned the comments below but these were not addressed in the discussion or addressed to a minor degree. Lines 283-287: The authors mentioned: “This suggests that girls are able to face conditions that challenge postural control and overcome them by executing slower (i.e., less accelerated) movements than boys.” What was the reason for this finding?
The authors have expanded the Discussion to answer your question based on previous articles:
Freitas, D.L.; Lausen, B.; Maia, J.A.; Gouveia, É.R.; Antunes, A.M.; Thomis, M.; Lefevre, J.; Malina, R.M. Skeletal maturation, fundamental motor skills, and motor performance in preschool children. Scan J Med Sci Sports 2018, 28, 2358-2368. doi: 10.1111/sms.13233
Remer, J.; Croteau-Chonka, E.; Dean, D.C.; D’Arpino, S.; Dirks, H.; Whiley, D.; Deoni, S.C.L. Quantifying cortical development in typically developing toddlers and young children, 1–6 years of age. Neuroimage 2017, 153, 246‐261. doi: 10.1016/j.neuroimage.2017.04.010
Malina, R.M.; Bouchard, C.; Bar‐Or, O. Growth, maturation, and physical activity, 2nd ed. Human Kinetics Publishers: Champaing, United States, 2004.
- Since the mean chronological age of the participants in this study was higher, could this be related to differences in developmental age?
This relationship is not supported theoretically because the difference is 0.2 years (less than two months old) or in practice (because the chronological age difference between the two groups is not statistically significant).
- Lines 291-294: A further and more detailed discussion is warranted.
That paragraph has been expanded:
“In the comparison between sexes, all the accelerometric variables of the tests in monopodal stance were statistically lower in the group of girls. This suggests that girls are able to face conditions that challenge postural control and overcome them by executing slower (i.e., less accelerated) movements than boys [23]. Although this finding may be due to the fact that the girls studied were in more advanced stages of maturational development. Prior studies have linked motor development in childhood to the growth of the central nervous system [42, 43] and have assumed that age and sex relationships among biological maturation and fundamental motor skills (like postural control) may be associated with changes in brain structure and underlying function neuromuscular maturation [44]. In parallel, Freitas et al. [42] indicate that taking into account the sex differences in biological maturation, skeletal maturation may be more closely related to neuromuscular maturation in girls than in boys at these young ages.”
Once again, thank you very much for the time spent and the interest shown in this work; as well as in the positive evaluations you have given of it.
Receive a warm greeting,
The authors.
Reviewer 2 Report
Dear authors
Thank you for this paper. I understand what your aims were and what you wanted to achieve but I don't think this is the correct design. A cross-sectional study doesn't give the direction of influence - that is, perhaps children are more active because they have better postural control, rather than physical activity causes better postural control?
I think it is difficult to trust the accuracy of the parents' reported PA for their children, unless it was during a school holiday. I find it difficult to believe that children of this age achieved 30 minutes of PA only on 2 - 3 days.
You have carried out a lot of t-tests and not adjusted for multiple testing. For example, the comparative analyses between children include: active girls vs sedentary girls; active boys vs sedentary boys; active boys vs active girls; sedentary boys vs sedentary girls (all carrried out four times under four different conditions - that's 64 t-tests alone). For these tests, a Bonferroni adjustment would give an adjusted alpha of 0.0008 as the value for statistical significance. I wonder why you didn't consider an ANOVA in the first instance.
The logistic regression results, although giving a significant p-value, suggests effect sizes (OR) that could well be negligible from a clinical viewpoint as opposed to a statistically-significant viewpoint. Would you consider that support for the potentially chance nature of these results is suggested by the contrasting outcomes e.g. higher accelerations for active boys and girls in some conditions? Combined with the limitation, which you recognise, that this is potentially an underpowered study (although you didn't carry out any power calculations), it is very difficult to place any trust in these results.
I am not sure for these reasons that your study adds anything,
Author Response
Dear Editor and Reviewer of Journal of Clinical Medicine:
Thank you very much for your suggestions and contributions to improve the quality of the manuscript. Following your indications, we respond, point by point, to the reviewers' comments.
In the text, all the modified or added sentences have been written in red to facilitate the correction by the reviewers.
- A cross-sectional study doesn't give the direction of influence - that is, perhaps children are more active because they have better postural control, rather than physical activity causes better postural control?
The reasoning he proposes is valid but, taking into account previous research and, with the present gender perspective, it is not supported. Throughout the Introduction section, the available evidence on the effect of physical activity on the development of fundamental motor skills is developed.
In parallel, the Discussion section has been expanded to reinforce the strength of our research and contextualize the findings obtained.On the other hand, the differences in the weight and height of boys and girls were not statistically relevant, for this reason it was not given relevance in the interpretation of the results.
- I think it is difficult to trust the accuracy of the parents' reported PA for their children, unless it was during a school holiday. I find it difficult to believe that children of this age achieved 30 minutes of PA only on 2 - 3 days.
The authors understand the criticism you make us. In fact, this is an aspect that the authors have included among the methodological limitations of this study (at the end of the Discussion section).
However, the authors have no solid reason to be suspicious of the veracity of the information transmitted by the parents. Furthermore, the levels of physical activity carried out by children, regardless of the trend in industrialized countries towards sedentary lifestyles, are highly varied between different countries and even between the different regions of each country.
- You have carried out a lot of t-tests and not adjusted for multiple testing. For example, the comparative analyses between children include: active girls vs sedentary girls; active boys vs sedentary boys; active boys vs active girls; sedentary boys vs sedentary girls (all carrried out four times under four different conditions - that's 64 t-tests alone). For these tests, a Bonferroni adjustment would give an adjusted alpha of 0.0008 as the value for statistical significance. I wonder why you didn't consider an ANOVA in the first instance.
Because, the most suitable test for the analysis of an independent variable in two independent sample groups is the two independent sample t-test.
The ANOVA, although it could also be applied, is more suitable for the comparison of more than two independent sample groups.
(This information is supported by “Choosing the Correct Statistical Test in SAS, Stata, SPSS and R” from UCLA: Statistical Consulting Group).
- The logistic regression results, although giving a significant p-value, suggests effect sizes (OR) that could well be negligible from a clinical viewpoint as opposed to a statistically-significant viewpoint. Would you consider that support for the potentially chance nature of these results is suggested by the contrasting outcomes e.g. higher accelerations for active boys and girls in some conditions?
The fact of having obtained statistically significant results in a small sample size suggests and supports that the relationships detected in this research are valid. However, as the authors recognize in the manuscript, this does not indicate that they can be extrapolated, at the moment, to the total population of children. At the same time, given that the results provide new specific associations (in relation to previous research), the authors have expanded the Discussion section to make it more robust.
As for your question, in no balance test did active boys and girls perform worse (that is, greater accelerations) than their sedentary counterparts. Only the active girls obtained higher accelerations than the sedentary girls during the NORMAL GAIT but, as it is a dynamic test, it is necessary to be cautious in its interpretation: the greater accelerations during walking are due, fundamentally, to higher walking speed and, This is consistent with the fact that this phenomenon is detected in the group that performs more physical activity.
Once again, thank you very much for the time spent and the interest shown in this work; as well as in the positive evaluations you have given of it.
Receive a warm greeting,
The authors.
Round 2
Reviewer 2 Report
Dear authors
Thanks for responding to my queries. I accept some of your answers.
Thank you for explaining to me what a t-test is for. However, you have still not explored the problem with conducting multiple t-tests. If you are simply comparing two independent samples, then use a t-test - just one.
But you carried out 64 t-tests: on active girls vs sedentary girls; active boys vs sedentary boys; active boys vs active girls; sedentary boys vs sedentary girls (all carried out four times under four different conditions - 64 t-tests. With an alpha of 5%, then the probability of making a mistake (Type I error) is 1 - (1 − 0.05)^64 = 96%. In other words, you are almost certain to have results that are Type I errors. This is why you correct for multiple testing or use an ANOVA. Perhaps “Choosing the Correct Statistical Test in SAS, Stata, SPSS and R” from UCLA: Statistical Consulting Group should have told you that an ANOVA is in fact a large number of t-tests but it also controls for errors so that the Type I error remains at 5% and any significant result is not just highly likely - e.g. 96% likely - to be wrong because you ran a lots of t-tests.
Author Response
Dear Editor and Reviewer of Journal of Clinical Medicine:
Thank you very much for your suggestions and contributions to improve the quality of the manuscript. Following your indications, we respond, point by point, to the reviewers' comments.
In the text, all the modified or added sentences have been written in red to facilitate the correction by the reviewers.
- Thank you for explaining to me what a t-test is for. However, you have still not explored the problem with conducting multiple t-tests. If you are simply comparing two independent samples, then use a t-test - just one. But you carried out 64 t-tests: on active girls vs sedentary girls; active boys vs sedentary boys; active boys vs active girls; sedentary boys vs sedentary girls (all carried out four times under four different conditions - 64 t-tests. With an alpha of 5%, then the probability of making a mistake (Type I error) is 1 - (1 − 0.05)^64 = 96%. In other words, you are almost certain to have results that are Type I errors. This is why you correct for multiple testing or use an ANOVA. Perhaps “Choosing the Correct Statistical Test in SAS, Stata, SPSS and R” from UCLA: Statistical Consulting Group should have told you that an ANOVA is in fact a large number of t-tests but it also controls for errors so that the Type I error remains at 5% and any significant result is not just highly likely - e.g. 96% likely - to be wrong because you ran a lots of t-tests.
Up to this point, the authors had not fully understood the correction you were making to us.
In this last message you have sent us, we have understood what you wanted to tell us.
The authors have modified the statistical analysis by an ANOVA test. Consequently, the level of significance of the differences detected has changed in some cases (Table 3). But, in general, the results have not undergone major changes.
Once again, thank you very much for the time spent and the interest shown in this work; as well as in the positive evaluations you have given of it.
Receive a warm greeting,
The authors.
This manuscript is a resubmission of an earlier submission. The following is a list of the peer review reports and author responses from that submission.
Round 1
Reviewer 1 Report
The present study aimed at investigate the potential link between physical activity and postural control in healthy children.
I have several concerns regarding this investigation and the manuscript.
This study has a fatal flaw in "evaluating" physical activity (PA). The authors "only" ask to the participants if they were “practicing PA or not and, if so, the number of days they did it”. Measuring PA cannot only based on "declaration", especially when this is the main hypothesis of the study. The period screened is also missing. The reader does know if this question was asked to the children or to their parents. In addition, physical activity is not defined. Without a clear definition, how a child or even a parent could answer to "do you practice or not". Later on in the manuscript, the reader “discover” that subgroups of sedentary or active participants were made without knowing how this was done. Nowadays several methods could help to objectively measure PA with validated questionnaires or even better with "real" accelerometric measures. Since the authors have used an accelerometer to assess postural control, it could have been used to assess PA, as well.
Another "small" concern, which is not small in fact. Do the authors believe that the accelerations values reported are the good ones? I was wondering how vertical acceleration values could be comprised between 13 and 60 g for “static” balance!!! ... especially with an accelerometer with a range of measure of +/- 6g.
Another major problem with the present investigation is the “unclear” rationale of the hypothesis formulated. The primary purpose of the Introduction is to develop the hypothesis that will be tested by the research design. Unfortunately, here, the introduction lacks, in part, of a clear argumentation on how the maturation process influence postural control and how physical activity will influence it as well (and physical activity from “when”?). In addition, there is no rationale related to possible difference between boys and girls. Hence, I do not “understand” how the authors could expect differences between sexes.
Line 74: why did you choose an age range just in the middle of hormonal changes? Did you assess biological age? Do you think this could influence your outcomes?
Line 82-84: not detailed enough
Line 99: density has no unit
Line 125: how “sedentary” and “active” were defined?
Line 117: why did the authors choose the maximal value? Is it representative of the “real” acceleration over the test? The RMS of the acceleration value is not defined
Table 1: significant difference should be reported alongside the value of boys or girls not next to the average value
Line 204: I do not understand how you could assume that the alignment is “better” with measures of accelerations, especially with only one accelerometer.
Although not clearly explained, having a lower acceleration means better postural control. Then how active participants could have higher acceleration than sedentary participants could?
Line 228: what are the factors that are conditioned by sex?
Line 232: then how could you process the data to verify this? Again I think that performing a test without measuring the duration of this test is a fatal flaw, especially if the aim was to compare boys and girls at an age were anthropometric changes are relevant.
Line 244: there are “girl” sports where lower limbs muscles are very well strengthen as well.
Line 247: girls less “active” but with lower acceleration….how?
Sometimes, figures are better than tables to have an overall view of the outcomes
Reviewer 2 Report
General comments
I appreciate the opportunity to provide comments on this manuscript, which sought to investigate the effect of PA on the postural control state of school children with two secondary objectives to identify which specific capacities of postural control benefited most from physical activity in children and to identify possible differences between the sexes. The paper is well written and I also find the thesis of this paper interesting and warranted but feel that one issue in particular needs to be addressed.
The authors have presented the relationship between the development of psychomotor skills and postural control. Given the chronological age of the participants, ~ 10 years of age, the authors should also mention physical developmental changes between the sexes and any effect that these differences may have on their findings. For example and per the anthropometric data that was presented in table 1, girls where taller and weighted more than their males counterparts. This finding is expected since the onset of puberty occurs at a younger age in females than males. This topic needs to be introduced and then addressed per the finding of this study.
Specific Comments:
- Title: The participants in the study are school children and the objective of the study is to assess PA on the postural control state in this population. Yet, the title does not imply it. Consider adjusting.
- Lines 40-47: I do not follow how this paragraph is related to the overall flow of the introduction. Clearly, PA is important to many aspects of normal development and growth. However, what is the relationship to the overall context of the paper and particular stability control? Please clarify.
- Line 54: The authors mentioned: “qualitative methods, which are inefficient and have low reliability” in regards to balance assessments. What about the use of balance plates. Please address and add.
- Lines 58-59: The authors mentioned: “The exact mechanism through which PA influences the development of alignment and balancing reactions in children……” The authors need to better define the study population. Specifically given the chronological age in relation to the developmental age.
- Lines 219-222: The authors mentioned: “This suggests that girls are able to face conditions that challenge postural control and overcome them by executing slower (i.e., less accelerated) movements than boys.” What was the reason for this finding? Since the mean chronological age of the participants in this study was higher, could this be related to differences in developmental age?
- Lines 226-228: The same as the comment above.
- Lines 268-277: Should the issue of chronological versus developmental age be considered a limitation to this study?